# Boys don't cry (or kiss or dance): A computational linguistic lens into gendered actions in film

**Victor R. Martinez**[1]*, **Krishna Somandepalli**[2], **Shrikanth Narayanan**[1,2]

**1** Department of Computer Science, University of Southern California, Los Angeles, California, United States of America, **2** Department of Electrical & Computer Engineering, University of Southern California, Los Angeles, California, United States of America

\* victorrm@usc.edu

## Abstract

Contemporary media is full of images that reflect traditional gender notions and stereotypes, some of which may perpetuate harmful gender representations. In an effort to highlight the occurrence of these adverse portrayals, researchers have proposed machine-learning methods to identify stereotypes in the language patterns found in character dialogues. However, not all of the harmful stereotypes are communicated just through dialogue. As a complementary approach, we present a large-scale machine-learning framework that automatically identifies character's actions from scene descriptions found in movie scripts. For this work, we collected 1.2+ million scene descriptions from 912 movie scripts, with more than 50 thousand actions and 20 thousand movie characters. Our framework allow us to study systematic gender differences in movie portrayals at a scale. We show this through a series of statistical analyses that highlight differences in gender portrayals. Our findings provide further evidence to claims from prior media studies including: (i) male characters display higher agency than female characters; (ii) female actors are more frequently the subject of gaze, and (iii) male characters are less likely to display affection. We hope that these data resources and findings help raise awareness on portrayals of character actions that reflect harmful gender stereotypes, and demonstrate novel possibilities for computational approaches in media analysis.

## Introduction

TV and film media often reflect the views of a society with a considerable impact in how gender stereotypes are created or reinforced [1]. Through assumed behaviors and social roles, media representation and portrayals are a major influence in the way we construct our beliefs and ideas around gender-appropriate behaviors and norms, particularly during the formative years of childhood and youth [2–5]. Even throughout our adulthood, these portrayals can still guide the way we think [6], how we create our worldview and perceptions of others [7], our fashion choices [8], and the perception of our self-identity [9].

**Data Availability Statement:** We have made the dataset and labeling models available for download at https://sail.usc.edu/∼ccmi/actions-agents-and-patients/.

**Funding:** The author(s) received no specific funding for this work.

**Competing interests:** The authors have declared that no competing interests exist.

Right now there is a clear disparity in the way characters are portrayed in TV and film media, particularly with respect to the characters' assumed or perceived gender expression. For example, over the past two decades, the media industry has been highlighted as one where women tend to be both underrepresented [10, 11] and depicted in a stereotypical manner [12, 13]. The former being so gravely unbalanced that male leads outnumber female leads two-to-one, with male characters speaking and appearing on the screen twice as often than their female counterparts [11]. Female characters are often presented in decorative (e.g., for their body and beauty), family-oriented, and demure roles [14–18]; male characters are typically shown as independent, authoritarian and professional agents and, unlike their female counterparts, these representations do not depend on the male actor's age or physical appearance [12, 19].

One of the main limitations in most of these efforts is their largely qualitative nature that require immense manual work with human annotations and/or surveys [20]. These cannot match the scale at which media content is currently being produced or consumed; in fact, these efforts have been unable to produce systematic data for both science and media scholarship at scale. To provide supporting evidence for systematic differences in gender portrayals at a larger scale, researchers have recently turned to machine learning models. These applications range from works that automatically detect actors' faces and voices in TV and film [21–24] to works on film narrative understanding through the analysis of character dialogues [25, 26]. A number of these are applied directly to movie scripts to gain insight into the early stages of content creation, and where the suggested modifications could be implemented at lower cost. For example, a linguistic analysis of gender ladenness (i.e., the degree of association in which language may be perceived as feminine or masculine [27]) in movie scripts found that romantic movies tend to include language with a higher degree of feminine association, whereas action movies tend to include language with higher degree of masculine association [28]. With respect to characters' dialogues, linguistic analysis studies have found that male characters are associated with a higher number of words related to achievement, whereas female characters are usually written with more positive language, lower agency, and less power than their male counterparts [26, 29]. Other types of studies center around the social network inferred from scene sharing among characters. These studies demonstrate that with a few exceptions, men play almost all central roles across all genres, and for every three characters interacting in a movie, at least two are men [25, 26].

These automatic approaches come with its own limitations. For example, even when there is a general concensus among researchers that gender lies on a spectrum, most of the automated media content tools are limited to identifying only the female–male dyad—a reduction that some might argue to be overly simplistic [30]. Another limiting factor is that gender stereotypes are not bound to dialogue or scene co-appearance, but also can be communicated through the actions and behaviors of the characters [31]. For instance, consider the following three common stereotypes. First, the *tomboy*, a stereotype embodied by girls who are interested in science, vehicle mechanics, sports or other gender non-conforming behaviors or appearances. Second, that of the *spinster* or *crazy cat lady*, an unmarried women of a certain age whose sole narrative arc centers around their quest to find a partner. Finally, the *scary angry man*, which often portrays persons of color as innately savage, animalistic, destructive, and criminal. These stereotypes are rarely described explicitly through the character's dialogues, and similarly one would be hard pressed to infer them through the character's scene co-appearance networks. Instead, audiences tend to infer the implied stereotypes through a complex understanding of the character's appearance, as well as their actions and behaviors [32, 33]. With this in mind, we believe there to be an open opportunity for a complementary analysis based on the character's actions to better understand the pervasiveness of harmful

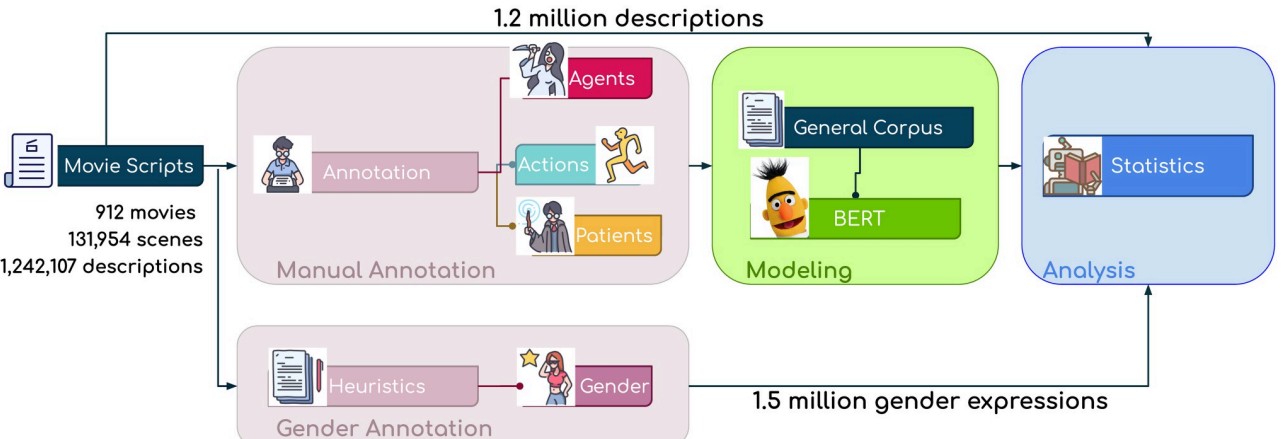

**Fig 1. Computational linguistic lens into gendered actions in film.** Our framework starts with a dataset of over 1.2 million movie descriptors from 912 movies and then implements three steps: first, the annotation process, where we collect manual annotations for 9,613 descriptions and over 1.5 million gender expression labels for characters. In the second step, we develop a machine learning model to identify actions, agents and patients from the natural language found in the movie scripts. Our model—trained on a large-set of general-domain documents and fine-tuned on a manually labeled description set—identifies actions, agents and patients from the natural language descriptions found in movie scripts. We use this model to automatically label the complete dataset for analysis. In the final third step, we perform a series of statistical analysis to uncover portrayal differences along characters' portrayed attributes.

gender representations in media. This work, to best of our knowledge, is the first to address this gap.

## Current research

In this work, we present a large-scale media content analysis framework to uncover differences in how characters engage in actions depending on their their assumed gender expression. To this end, we perform three steps: (i) data collection and annotation; (ii) machine-learning modeling, and (iii) statistical analysis. Fig 1 presents an overview of the complete process. In the following we will briefly describe each of these steps.

## Data collection and annotation

We start from a collection of 912 movie scripts from which we extracted 131,954 scenes with 1,242,107 action descriptors (i.e., sentences that describe an action occurring during a scene). We design a task to collect manual annotations for a small sample of these action descriptions. This provides us with a manually labeled dataset for actions and their constituents (i.e., the agents and patients engaged in the action). Additionally, to identify the assumed gender expressions for each character in the movie scripts, we design a heuristic-based method based on identifying usage of proper names, gender pronouns and manually labeled examples.

## Machine-learning modeling

While the previous step can provide reliable labels, it can only do so for a small subset of the action descriptions. To scale our approach, we develop a state-of-the-art machine-learning model that identifies characters engaged in actions from the natural language descriptors found in the movie scripts. Our approach leverages transformers, particularly the recent developments in large-scale contextual language models (BERT) [34]. We fine-tune the pre-trained general-knowledge BERT model to allow the model to learn the different ways in which

scriptwriters describe character actions and behaviors. We use the model to automatically label the remainder of the descriptions for actions, as well as the characters acting as agents and patients of the action. This process yields a set of 1.2M+ automatically labeled descriptors, containing over 50,000 different actions with over 20,000 different participants.

## Statistical analysis

As a final part of this work, we show how this framework can uncover portrayal differences along character attributes such as gender. To this end, we perform a series of statistical analyses over the 1.2+ million action descriptions to estimate the frequency of portrayal as a function of role and gender. As part of our results, we provide insights into some of the nuanced aspects in which certain stereotypes are being communicated through character actions and behaviors. Based on previous literature, we categorize our findings on the differences in portrayals into three main groups: (i) in how female characters are often depicted with less agency than male characters [29]; (ii) on the emphasis placed on the female appearance and sexual objectification of women actors [35, 36]; and (iii) on how gender plays a role in the frequency of affective portrayals, either by frequently casting women into overly emotional roles [12, 37, 38] or by a clear absence of male portrayals of affection [39, 40].

These analyses, and the insights obtained, exemplify how the computational framework can be beneficial for film makers, scriptwriters, and the broader society in terms of creating awareness about media portrayals from an equity lens.

In summary, the contributions of this work include:

1. the design, construction and public sharing of an action description dataset with over 1.2 million descriptions obtained from 912 movie scripts

2. the proposal for a machine learning method to identify actions, agents and patients from linguistic cues found in the action descriptions of a screenplay, and

3. Implementation of a statistical framework with the 1.2 million action descriptions to highlight biases in the gendered portrayals of actions, agents and patients in media

To provide other researchers with the opportunity to explore additional hypotheses, we have made the dataset and labeling models available for download at https://sail.usc.edu/~ccmi/actions-agents-and-patients/.

## Data collection and annotation

In this section we describe the process used to obtain manual annotations for actions and the characters engaged in these actions as agents or patients. Furthermore, we describe the heuristics used to identify the character's assumed gender expressions from the language used throughout the movie script.

Our analysis is performed over a dataset of 1.2 million action descriptions obtained from a publicly available collection of 912 movie scripts covering over 31 genres and 104 years of movie productions (1909–2013) [41]. This collection of movie scripts has been widely accepted and validated by the research community, particularly in the analysis of character portrayals [29, 42]. Thus, we believe this corpus provides a representative sample of the typical character actions and behaviors on screen. Moreover, it is one of the few resources publicly available to provide human-validated named entity recognition, co-reference resolution, and gender information for each of the main characters in each movie script.

**Table 1. Descriptive statistics for the action description dataset.**

| | Min | Mean | Median | Max | Std. |
|---|---|---|---|---|---|
| Movies: 912 | | | | | |
| Genres: 31 | | | | | |
| *Per movie* | 2 | 3.00 | 3.00 | 7 | 0.95 |
| Scenes: 131,954 | | | | | |
| *Per movie* | 1 | 1363.45 | 136.00 | 646 | 68.52 |
| Action Descriptions: 1,242,107 | | | | | |
| *Per movie* | 1 | 1363.45 | 1313.00 | 4901 | 560.03 |
| Actions: 1,634,230 | | | | | |
| *Per movie* | 1 | 1797.83 | 1765.00 | 4057 | 633.98 |
| *Per action description* | 0 | 1.32 | 1.00 | 16 | 1.06 |
| Agents: 1,175,237 | | | | | |
| *Per movie* | 1 | 1008.93 | 977.00 | 3061 | 361.33 |
| *Per action description* | 0 | 0.95 | 1.00 | 11 | 0.56 |
| Patients: 960,970 | | | | | |
| *Per movie* | 2 | 1056.01 | 1000.50 | 4286 | 464.94 |
| *Per action description* | 0 | 0.77 | 1.00 | 14 | 0.89 |

## Action description dataset

From each movie script, we collect all of its action descriptions. Action descriptions are paragraphs within a scene that tells the reader what is going to happen on the screen and describes the characters and their actions. We focus on these descriptions as a complementary approach to prior works based on character dialogue [25, 26, 29]. We split each action paragraph into sentences and identify the actions (verbs) using Spacy [43]. This process yields a total of 1, 242, 107 sentences ($\mu = 1363.45$, $\sigma = 560.03$, $M = 1313$ per description), 1, 634, 230 predicates and 84, 513 unique actions. Table 1 presents descriptive statistics of the constructed dataset.

## Manual annotation

We select a sample of 12, 500 sentences to be coded by human annotators. This sample is constructed by rejection sampling, where each sentence has to have at least one verb (action). Our annotation procedure consisted of two tasks: labeling and verification. The following sections describe each task in detail.

A total of 981 annotators were hired through Mechanical Turk. From these, 602 were assigned to the labelling task, and 379 to the verification tasks. On average, each annotator took around a minute to complete an annotation task ($\mu = 50.21s$; $\sigma = 173.69$), while only requiring half the time to verify an annotated result ($\mu = 28.49s$; $\sigma = 30.78$). Remuneration scheme was devised to ensure that the annotators receive at least an hourly minimum wage in the U.S. This was calculated by diving the current (2021) minimum wage per hour ($7.25) by the expected time it would take to complete a single annotation ($\sim 1$ minute).

**Labeling.** For the labeling task, we present non-expert annotators with a sentence and ask them to identify the agents (or patients) for a particular action (verb). To ensure that annotators label only characters (and not inanimate objects), our annotation style follows the work of [44] where the syntactic heads of the constituents are annotated instead of their full extent. Moreover, we simplify the annotator's task by providing annotators with two pieces of information: a pre-selected action, and a list of possible entities (see Fig 2). The former is obtained from the part-of-speech tags provided by the corpus. For each verb, we created a separate

**Fig 2. Labeling task.** Annotators are presented with a sentence and an action. They are asked to either select the agent (source) and patient (target) of the action. For cases where one of these is missing, the annotator has the option to check the 'Does not say' box.

annotation task where annotators identify the agents and patients for that particular action. For the latter, however, we were not able to provide a list of all possible characters, as accurately identifying the literary figures in a text is still an open research problem [45]. Instead, we provided our best attempt at a reduced list of possible characters by identifying entities that are more likely to be used as character names. To construct this list, we start by filtering out words that are not pronouns, proper nouns, nouns, or noun phrases. From the remainder, we remove most of the common words since these are not normally used to refer to a character (e.g., door, eyes, room, hand, car, head, floor, etc. . .). For the special case of honorifics (such as Mr. or Miss), we follow [45] in considering these tokens as part of an entire maximal span (e.g., [Mr. Collins] or [Miss Havisham]). We include this maximal span as a single entry in our select box. For cases where the agent (or patient) is not explicitly stated in the sentence, annotators have the option to check in the box for 'Does not say'.

Each sentence was annotated by 3 non-experts and their agreement is used as the presumptive label for the next stage. If there was no agreement between the annotators, we discarded the sentence from the sample.

**Verification.** In this stage, we ask another annotator to verify if these labels are correct or not. We present a single sentence, the pre-selected action, and the presumptive labels for agents and patients when applicable or a 'Does not say' string for when not. The annotator gets prompted with a single question: "Is [AGENT] doing the [ACTION] to [PATIENT]?" and radio buttons for Yes, No or Does not say. If the annotator does not agree with the result, or if they cannot say whether it is correct or not, we discard the particular sentence from consideration.

As a final quality control step, one of the authors checked all the sentences and verified the labels for consistency. We decided to follow this multi-tiered approach to ensure a high level of annotation quality, one that makes sure there is a marked distinction between *subjects* and *patients*. This distinction becomes paramount if one considers that the models we will be working with do not have an inherent notion of what a character is. Hence, if the annotation results in low-quality labels, we run the risk of having a model that picks any word from the predicate as the patient for a given action. As we will discuss in further sections, our two-step verification was needed to ensure that most of the labels were correct and identify unreliable annotators.

## Inferring assumed gender for characters

To obtain a character's assumed gender expression, we follow a hierarchical heuristic approach. This approach is heavily informed by prior work on the same domain [25, 26, 29]. Our gender estimation method proceeds as follows: first, for movie scripts of an already produced film, we obtain the character's gender from the casting of that role from IMDb http://imdb.com. For the remainder of the characters, we rely on the following heuristics:

1. For proper names, we estimate gender using historical U.S. census data [46].

2. We use gendered pronouns as markers of that character's gender. The set of pronouns was selected from Twenge et al. [47], and includes the following words: female pronouns (e.g., she, hers, her, and herself); male pronouns (e.g., he, his, him, and himself), and neutral (or plural) pronouns (e.g., we, they, them).

3. Lookup over a manually collected word list containing the 3,000 most frequent words and their gender. This list includes gendered words for family and relationships (e.g., uncle, aunt, wife, husband), common gendered nouns and other words where gender is presumed evident (e.g., boy, gal, policeman, congresswoman).

From the total of $n = 2,136,207$ character instances, our hierarchical heuristic method is able to infer the gender for 71.64% of instances ($n = 1,530,587$). From these, 917,114 correspond to agents and 613,473 to patients (see Table 2). In line with previous research, the sample of genders contains male characters in a 2-to-1 proportion to female characters [10, 48]. We manually identify 49,885 cases of character references which should have a gender, but our heuristics were not able to determine which gender it should be. A majority of these cases (56.4%) are unresolved co-references (e.g., I, you, me), and could be addressed in future work when appropriate literary co-reference systems become widely available.

**Gender estimation performance.** We estimate the performance of our gender estimation through a manual verification process. This process involved a manual inspection of the dataset to collect 400 character names alongside their gender (i.e., non-gendered, male, female, and neutral). We then verify that our heuristics are able to infer the correct gender for each of these instances as a classification task. We report classification performance as part of our results.

## Machine-learning model

In this section we provide an overview of the computational model that identifies the action and its constituents (agents and patients). This model is based on the current state-of-the-art BERT-based models for semantic role labeling (SRL) [49]. Additionally, we present the steps performed for domain-adaptation, which results in a significant improvement in performance over competitive baselines.

**Table 2. Gender distribution of agents and patients.** Our sample displays a 2-to-1 ratio of male to female characters. Proportion tests reveal an unequal proportion of Male agents to Female agents, and Female patients to Male patients ($p < 0.0001$).

|         | Female  | Neutral | Male    | Unknown | Total     |
|---------|---------|---------|---------|---------|-----------|
| Agent   | 227,196 | 176,596 | 479,136 | 34,186  | 917,114   |
| Patient | 144,890 | 168,092 | 284,757 | 15,734  | 613,473   |
| Total   | 372,086 | 344,688 | 763,893 | 49,920  | 1,530,587 |

## Automatic identification of character actions

We frame the problem of automatically identifying the set of actions and its participating characters as a Semantic Role Labeling task (SRL), with a few differences. Given a sentence, the SRL task consists of analyzing the propositions expressed by some target verbs of the sentence. In particular, for each target verb, all constituents in the sentence which fill a semantic role of the verb have to be recognized. Typical Semantic arguments include Agent, Patient, Instrument, etc. and also adjuncts such as Locative, Temporal, Manner, Cause, etc. [50].

According to Shi et al. [49], a typical formulation of the SRL task is split into four subtasks: predicate detection, predicate sense disambiguation, argument identification, and argument classification. We start from the assumption that there are models that support the first two tasks in a reliable manner. Specifically, in our experiments, we use Spacy [43] for predicate identification, which allows us to focus entirely on the argument identification and classification sub-tasks. Furthermore, in contrast to the traditional SRL task [50], we are only interested in the characters performing the action, and those that are the object of an action. Hence, our label set can be restricted to actions, agents, and patients only. Another point of contrast is that we explicitly make the distinction between objects (inanimate) and patients (characters).

## Proposed model

We follow Shi et al. [49] in applying a simple yet powerful recipe for SRL: obtain word vector representations from a pretrained BERT-based architecture to train a Recurrent Neural Network for sequence labeling. Our proposed model (see Fig 3) learns to map the sequence of tokens from an input sentence to a sequence of labels for actions, agents, and patients. Inputs to our model are sentences, which are tokenized and fed into a BERT model to obtain highly contextualized word representations. These representations are then used as input to a Recurrent Neural Network to produce a sequence of token-level labels.

In contrast to Shi et al. [49], who inputs the predicate as an additional feature to the model, our current setup restricts role labeling to a single predicate per sentence. If a sentence has more than one predicate, we create a separate copy for each predicate; this same setting was applied by Daza and Frank [51] and Zhou et al. [52]. In the following sections, we provide further details into the steps taken by our model.

**Input representation.**   We selected a BERT model for our input representation because of its remarkable success on a variety of NLP tasks, such as question answering, dialogue systems, and information extraction [53]. In this work, we start from the original BERT model (https://github.com/google-research/bert) trained for the general-domain on a large unlabeled plain text corpus—that is, the complete English Wikipedia and BookCorpus.

We follow the traditional format used for sentence encoding in the BERT transformer [53]. To obtain an input representation, we feed an action description into the model as an input sequence. This sequence starts with a sentence delimiter ([CLS]) and ends in a separator delimiter ([SEP]), as follows:

$$[CLS] \; word_1 \; word_2 \; word_3 \; \ldots \; word_k \; [SEP]$$

Our first step is to tokenize the sentence elements using WordPiece [54]. The WordPiece tokens are fed into pre-trained BERT models from which we obtain one vector representation for each of the tokens. Formally, the sentence representation step maps a delimited sequence

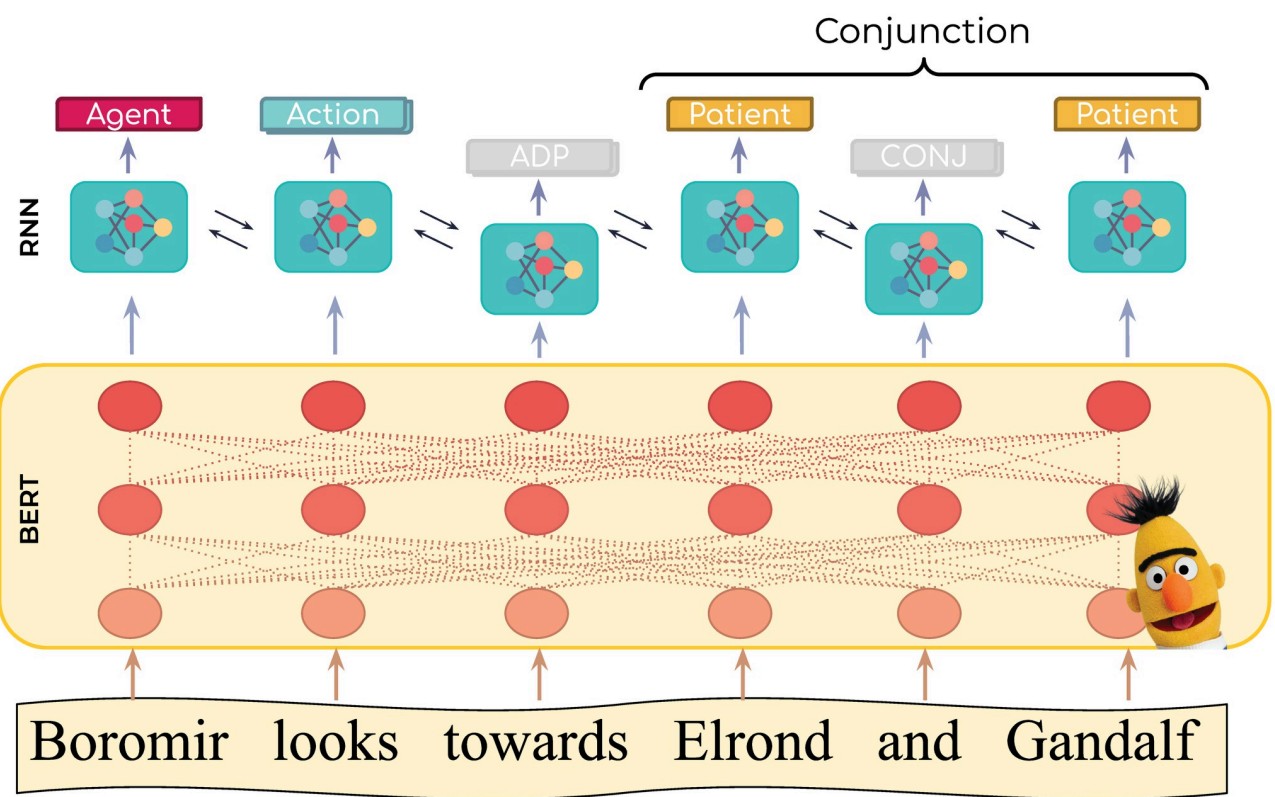

**Fig 3. Proposed SRL system.** Starting at the bottom, the input to the system is an action description in natural language. The output, shown at the top of the figure, is a sequence of labels (one per word). Labels indicate whether this word depicts an action, agent, patient or none. From its inputs, our model obtains a highly-contextualized representation for each word using the BERT transformer [53]. Each representation corresponds to a high dimensional dense vector that encodes the semantics of that word and the context it plays within the sentence. The sequence of vector representations is then fed into a recurrent neural network and a softmax layer for sequence labeling. As a post-processing step, a set of heuristics aggregate conjunctions to handle the case of groups of agents or patients.

of $k$ words into $n$ WordPiece tokens, as follows

$$[CLS], w_1, w_2, \ldots, w_k, [SEP] \rightarrow t_{[CLS]}, t_1, t_2, \ldots, t_n, t_{[SEP]} \qquad (1)$$

Note that the length of these sequences might not be the same as the tokenizer might split a word into multiple sub-word tokens. The token sequence is then mapped into the representation sequence by the BERT model. Let $H$ denote the sequence of BERT representations, given by

$$H = \{h_{[CLS]}, h_1, h_2, \ldots, h_n, h_{[SEP]}\} \qquad (2)$$

The dimension of each $h_i$ is given by the BERT model, and corresponds to 768 and 1024 for the bert-base and bert-large models, respectively. The sequence of highly-contextualized vector representations is then fed into a RNN for token-level prediction.

**Token-level prediction.** We use a bidirectional recurrent neural network (RNN) to obtain the semantic role labels for each word in a sentence. RNNs are a class of neural networks specialized in processing sequences of inputs. With each input, the RNN updates its internal state (memory) and produces a probability distribution over the labels for that input. Here, we feed the output of BERT (Eq 2) into the RNN layer as sequence of tokens, for which we obtain a

sequence of probability distributions. To obtain the sequence of SRL labels (i.e., Verb, Agent, and Patient), each token gets assigned to the label with the maximum (posterior-)probability. For the RNN, we explore two popular configurations: Long short-term memory cells (LSTM) [55] and Gated Recurrent Units (GRU) [56].

Formally, the sequence obtained from BERT, $H$, is then fed into a bidirectional RNN to learn a mapping between the tokens and the SRL labels of interest. The RNN takes the sequence of representations $H$ and outputs a sequence of $n$ hidden vectors $\{v_{[CLS]}, v_1, \ldots, v_n, v_{[SEP]}\}$. Each $v_i$ is constructed as the concatenation of the left-side and right-side context, $v_i = [\overrightarrow{v_i}; \overleftarrow{v_i}]$. To predict a label, we use a fully connected dense layer and a softmax function over all labels:

$$\begin{aligned} s_i &= \phi(v_i) \\ \hat{y} &= softmax(s_i) \end{aligned} \qquad (3)$$

where $\phi$ is the activation function. In our experiments, this function corresponds to a linear activation function $\phi(x) = xA^\top + b$ where $A$ and $b$ are learn-able parameters of the model. Finally, the complete model is trained using a weighted cross entropy loss where $w_C$ is the weighted associated to the class $C$

$$\mathcal{L}(y, \hat{y}) = w_C(-y[C] + \log\left(\sum e^{\hat{y}}\right))$$

**Post-processing.**   From the SRL system, we collect two outputs: the subword WordPiece token representation of the sentence, and the sequence of token-level labels. To recover the word-level sentence, we post-process the outputs of the RNN (Eq 3) by removing the special tokens introduced by BERT (i.e., [CLS], [SEP], [PAD] and [MASK]) and merging back Word-Piece tokens into words. The word-level label is calculated from its corresponding tokens as the mode of the token's label. Additionally, we postprocess the data further to accommodate for cases where agents and patients are composed of more than one word. Examples of this include the case of honorifics (e.g., 'Mr. Anderson', 'Captain Crunch') or expressions with more than one character (e.g., 'Mr. Smith and his wife'). We restructure our word sequence by (i) concatenating consecutive words with the same label into a single term, and (ii) merging consecutive terms that end up in a conjunction. The labels for word groups is assigned to be equal to the label of the left-most word. For example, after applying our model and post-processing procedure to the sentence presented in Fig 3 we are able to discern that "Boromir" is the agent of the action "look", and that the patients correspond to "Elron and Galdalf".

## Fine-tuning

Fine-tuning aims to adjust the BERT language model and its vocabulary for the way language is used in a particular domain, without offsetting the generalization power of the original model. This procedure typically results in models that achieve state-of-the-art performance for domain-related tasks. In this work, we perform fine-tuning of BERT language models by continuing the train of the model end-to-end on the action description dataset described in the previous section. In end-to-end training, we back-propagate the errors from the sequence labeling task back through the network, all through the BERT transformer layers (yellow box in Fig 3). This results in the update of the BERT layer parameters, which adapts them to the patterns in the language of the movie scripts.

**Model performance comparison.**   Model performance was estimated over the manual annotated dataset of $n = 9,613$ action descriptions. For training, we used 75% of the available

data ($n = 7, 209$). Performance was estimated on 15% of the data, held back as a test set ($n = 1, 419$). A development set (10%, $n = 985$) was used for parameter optimization and early stopping. Our experiments measure the model's ability to correctly categorize each token by its corresponding semantic role label. Hence, this can be seen as a 4-way classification task (i.e., Action, Agent, Patient, and None). We report the average accuracy and micro-average F-score for this classification task.

**Baselines.**   We compare the performance of our proposed approach to 3 baseline models, which were selected informed by previous literature in the SRL task. Our first baseline corresponds to a named-entity based approach proposed by Sap et al. [29] where we leverage part-of-speech tags and syntactic dependency tress to identify actions, Agents and Patients. The second and third baselines follow state-of-the-art BERT-based models for semantic role labeling (SRL): SimpleBERT [49] and AllenNLP SRL [57]. Even though our architectures are quite similar, there are a few differences between our approaches, particularly with respect to the inputs and the sequence labeling layer. A first difference is that AllenNLP uses a time-distributed linear dense layer to classify the sequence of outputs from the BERT system into the sequence of semantic role labels. In contrast, SimpleBERT and our method use a single RNN layer, as this might be better adapted to handling sequence data. Second, both SimpleBERT and AllenNLP extend the sentence representation of BERT to include the current predicate, as a way of informing the model which action to attend to. Instead, we restrict our inputs to only a single predicate per sentence. Finally, we note that SimpleBERT and AllenNLP are trained on non-literary data. Hence, these models serve as a direct comparison to the performance of out-of-the-box state-of-the-art SRL systems when applied to a novel real-world application.

We control for this possible source of noise by presenting two versions of each baseline. In the first version, the models are only provided with the action description and have to perform predicate identification and semantic role labeling. In a second version, we provide the sentence and the gold standard predicate, so the models only need to do the semantic role labeling task. We refer to the second version as the oracle predicate version.

## Statistical analysis

We propose a statistical model to identify significant differences in the frequency of the action portrayals due to the role and gender of its participants. We do this through a series of four studies. These studies explain the response variable (action frequency) as a function of different independent variables. The first and second studies examine the relation between action frequency and agent's or patient's gender independently. We denote the results from this studies by (**S1: agent-only**) and (**S2: patient-only**), respectively. The third study investigates the effects of the agent's and patient's gender simultaneously by including an interaction term as part of the co-variates. This model highlights actions that occur more frequently only when agents' and patients' have a particular gender each. We denote this model by (**S3: agent & patient**). Finally, our fourth model studies the dependency between action portrayals dependent on film trends over the years. We denote this study by (**S4: agent & patient + time**).

In each study, a generalized mixed-effects linear regression model is used to relate the genders and roles to the frequency of portrayals. We describe our statistical formulation in the following sections.

## Regression models

We part from the assumption that if there is no difference between the number of times an action is portrayed with respect to any particular gender, then this action does not reflect a

gender stereotype. This gives us a natural framework for posing the problem of identifying stereotypes as a regression over the frequency of actions and the gender of its participants.

To uncover this relation, we use a Poisson-regression generalized linear mixed model (GLMM; [58]). GLMMs are an extension of generalized linear models (e.g., logistic regression) to include both fixed and random effects. A fixed effects approach was chosen over a random-effects approach because our data contain repeated measurements (i.e., a single character can participate in multiple actions, multiple times). Furthermore, we use a Poisson regression as it is particularly useful for response variables that represent counts or frequencies [59].

The formal specification of the GLMM is as follows. Let $Y = [y_1, y_2, \ldots, y_N]$ be our response variable. Each $y_i$ corresponds to the number of times we see action $i$ in our dataset. We assume that $Y$ follows a multi-variate Poisson distribution, and we model the expected value as a linear combination of unknown parameters,

$$E[Y \mid \gamma] \approx \eta$$
$$g(\eta) = X\beta + Z\gamma$$

(4)

where $X$ and $\beta$ are the fixed effects design matrix, and fixed effects; $Z$ and $\gamma$ are the random effects design matrix and random effects. The link function $g$ corresponds to the log-link function log. We vary our predictor variables $X$ and random effects matrix $Z$ according to the each study as we describe in the forthcoming sections.

**Parameter estimation.** Regression parameters are estimated by deviance minimization, a generalization of the idea of using the sum of squares of residuals in ordinary least squares to cases where model-fitting is achieved by maximum likelihood [58]. Additionally, we identify statistically significant coefficients through a series of hypothesis tests on the coefficients. For each coefficient, we perform a Z-test and correct for multiple comparisons using Holm-Bonferroni method.

**Model validation.** For each of our presented studies, we validate its corresponding regression models by comparing the performance with two reduced models: *null* and the *no-interaction* models. In the null model, no additional explanatory variables are used. This model seeks to explain frequency of portrayal as a function solely of the control variables $Z$. In the no-interaction model, we remove the random effect–cross–gender interaction variable. By comparing against these two reduced models, we can provide statistical evidence for the existence of a relation between the gender of the characters ($X$) and the frequency of portrayals ($Y$). Furthermore, by comparing against the no-interaction model, we provide evidence for the hypothesis that the interaction between the agent's (or patient's) gender and actions makes for a better predictor than just the action and gender on their own. All model comparisons were done using a likelihood ratio test ($\chi^2$ test) at a significance level of $\alpha = 0.05$.

## Studies

**Study 1: Agent-only.** For our first study, the fixed effect matrix, $X$, corresponds to an indicator variable of the gender of the agent. Genders are encoded as $M$, $F$ and $N$ for male, female and neutral, respectively.

Additionally, we control for two sources of variability as part of the random effect matrix, $Z$. The first effect controls for the distribution of actions. That is, the fact that naturally some actions will occur more often than others. For this, $Z$ incorporates actions as a categorical covariate ($\mathbb{C}_a$). The second effect we control for is the movie genre. This comes from the fact that certain actions are more common than others for a particular type of movie. For example, we can expect the action 'run' to be more common in action movies than in dramas and romantic comedies. We obtain the genres for all of the movies in our data set from IMDb, and transform

them into a binary matrix, $m_g \in \{0, 1\}^{N \times G}$, where each entry indicates whether that movie has that particular genre. We include $m_g$ as an additional co-variate in our model. Formally, $Z$ is given by

$$Z = [\mathbb{C}_a; m_g] \tag{5}$$

To fit this study's regression models, we sub-sample the action description dataset to include only those records with a known agent. We observe that the gender distributions of this sub-sample still follows the gender distribution of the full dataset. Moreover, it also follows previous reports in the literature, that is, it remains as a 2-to-1 male to female ratio [10, 48].

**Study 2: Patient-only.**   Similarly to our first study, in the second study, the fixed effect matrix, $X$, encodes an indicator variable of the gender of the patient. It also uses the same random effect matrix as the previous study (Eq 5)

To fit the regression model, we sub-sample the action description dataset to include only those records with a known patient. This sub-sample also follows the gender distribution of the full dataset and the 2-to-1 male to female ratio.

**Study 3: Agent & patient.**   For the third study, the fixed effect matrix, $X$, is designed to reflect the gender group of agents and patients. This gender group is constructed as a categorical value for the combinations of male-to-male, male-to-female, female-to-male, and female-to-female. We use the same random effect matrix, $Z$, as in the first study (Eq 5).

For this study, the regression model is fitted to those records for which we know the gender of both agent and patient.

**Study 4: Agent & patient + time.**   It could be argued that the frequency of actions can be explained by changes in film trends over the years. Since our sample already provides a resource that covers a large span of production years (1909–2013), in our final study, we investigate if the frequency of portrayals can be explained as a function of both time and character demographics. To this end, we include an additional control variable $m_y \in \{1909, 2013\}^N$, the year the movie was released, as part of the control variables $Z$. By following this approach (as opposed to just including the variable as an additional co-factor), we ensure that the model learns that different years might follow different trends. The final control effect variable $\tilde{Z}$ is given by

$$\tilde{Z} = [\mathbb{C}_a; m_g; m_y]$$

## Results

### Annotation verification

**Manual annotations.**   From the 12, 500 sampled sentences, we found 14, 344 actions to be annotated. Annotators agreed with both the agents and patients for a large majority of the cases $n = 11, 775(82.09\%)$. Although, in one out of five, the verification annotation considered their answers to be incorrect ($n = 2, 162(18.36\%)$). A posterior analysis on the samples that were deemed incorrect reveals that most (90%) of the errors were caused by a small subset of annotators. We believe this annotators skipped the task completely by marking the 'Does not say' box and submitting. After discarding the incorrectly labeled samples, we are left with a dataset of 9, 613 sentences with at least one action.

**Character's gender estimation performance.**   Table 2 presents the gender distribution of our dataset. Previous works argue that male characters are generally given more agency than female characters [11, 29]. We are able to corroborate this finding through proportion tests. Our results reveal that the proportion of male agents is significantly higher than that of female

**Table 3. Gender classification report for the proposed heuristics over 400 manually annotated samples.**

|         | Precision | Recall | F1-score |
|---------|-----------|--------|----------|
| Female  | 91.0      | 67.0   | 77.0     |
| Male    | 91.0      | 63.0   | 75.0     |
| Neutral | 95.0      | 80.0   | 87.0     |
| Average | 92.33     | 70.0   | 79.66    |

agents ($Z = 294.12$, $p < 0.0001$). Additionally, the proportion of female patients is larger than that of male patients ($Z = 294.12$, $p < 0.0001$).

The results of the gender estimation heuristics performance on a set of 400 manually selected samples is shown in Table 3. This table reports precision, recall and F1 for each one of the gender groups, as well as average metrics across all groups. Our method achieves 75% accuracy and macro-F1 score of 77.75%, with individual F1 scores rating from 72.0% (non-gendered) to 87.0% (neutral). From their class-level results, we observe that our gender classification method is fairly precise when labeling female and male characters as well as gender-neutral terms (e.g., they, them).

## Machine-learning model performance

The complete system-level performance results are presented as part of the S1 Table. With respect to identifying agents and patients, the baseline models achieve high precision but suffer from low recall. While a high precision is important, we must consider that this model is to be used in the context of a large-scale analysis of characters and their actions. To obtain the most representative sample for such analysis, we would want our model to retrieve as many instances of actions and their participants. Hence, models with a higher recall ought to be preferred.

Even though there is a clear domain mismatch in the way the baselines were trained (news wires vs. movie scripts), both baselines can still recover some of the signal present in the dataset. In contrast, our naïve approach of relying on a pre-trained model BERT-base (uncased) resulted in no patient label being produced, and thus a 0.00% F1 score for that category. This suggests that movie scripts, and character names specifically, always follow a proper grammar.

Our results show that the domain adaptation of BERT language model resulted in the biggest improvement overall. For example, domain adapting SimpleBERT [49] resulted in a 6 percent increment in action identification, and about 3 to 4% (absolute) gains in agent and patient classification. Furthermore, our proposed model, trained end-to-end, achieved over 30% (absolute) above the baseline performance. The best performing model was our proposed conjunction of transformer and RNN (GRU), where the transformer was initialized with a BERT-base cased pre-trained model, and the complete set of parameters was updated end-to-end. This model achieves F1 scores of 96.80, 89.78 and 73.00 percent for action, agent and patient respectively. Compared to the baseline models, the difference in these performances was found to be significant (permutation test, $n = 10^5$, all $p < 0.05$). Surprisingly, even though we did not precondition BERT with the current predicate (as both SimpleBERT and AllenNLP do), our model was able to correctly infer the action for most of the sentences.

Finally, we investigated changes in the performance of the proposed model due to different RNN dimensions (see S2 Table). The model seems to get saturated around a dimension of 300, with higher dimensions not performing particularly differently from the current size. This result also suggests that the poor performance of the SRL baselines could be due to their larger size.

## Statistical models results

Across all four studies, models that include gender predictors performed significantly better than the null models ($\chi^2$ tests, all $p < 0.0001$). Including interactions between gender and action improved the performance of all models ($\chi^2$ tests, all $p < 0.0001$). These results support our assumption that gender plays an important role in defining the frequency of portrayals for a particular set of actions.

**Study 1: Agent-only.**   The regression model with gender as a co-variate achieved significantly lower AIC than the null model ($\chi^2(3) = 17, 375, p < 2.2e - 16$). Furthermore, including the interaction variable between action and agent's gender reduces AIC significantly further ($\chi^2(5723) = 15, 311, p < 2.2e - 16$). From the fitted regression models, we identify 378 actions for which an agent's gender plays a significant role in the frequency of the action portrayal (t-test, $\alpha = 0.05$). S2 Table shows the list of significant regressors. It is important to note that some of the identified actions appear to contain errors. For example, errors in the lemmatizer or parsing modules (e.g., confusing the past tense 'lie' with 'lay'), intransitive verbs (e.g., 'dress', 'wear'), or verbs whose subject is a thing and not a movie character. We have manually identified these errors and color coded them in our results.

**Study 2: Patient-only.**   Once again the regression model with gender as a co-variate showed a significantly lower AIC than the null model ($\chi^2(3) = 17543, p < 2.2e - 16$). Including the interaction term between patient's gender and action lead to even lower AIC ($\chi^2(5723) = 6456, p < 0.0001$). Similarly, we identify 60 actions for which a patient's gender plays a significant role in the frequency of the action portrayal (t-test, $\alpha = 0.05$). S3 Table shows the list of significant regressors.

**Study 3: Agent & patient.**   The regression model with gender groups achieved a significantly lower AIC than the null model ($\chi^2(3) = 16789, p < 2.2e - 16$). Furthermore, we found that a model that considers action-gender interactions provides a significant better explanation to the frequency of portrayals than the model without interactions ($\chi^2(5723) = 8951, p < 2.2e - 16$). We identify 135 instances where the frequency of an action is significantly impacted by the genders of the agents and patients portraying those actions. S4 Table show these significant relations.

**Study 4: Agent & patient + time.**   For our last study we see a similar trend, even when controlling for year, gender co-variates provide a significant reduction in AIC ($\chi^2(3) = 18751, p < 2.2e - 16$). Further decreased by incorporating the action–gender interaction term ($\chi^2(5723) = 6933, p < 2.2e - 16$). However, out of the four studies, this model only produces a handful of significant results. The reduced number of results we obtained for this study could be due to only a few actions happening consistently across the span of several decades. If an action only occurs a few times in a given year, our model might not be able to pick this difference up due to issues with statistical power. Hence the need for the aggregated results from studies 1 to 3. From the actions that do happen consistently across decades, we identify 23 actions for which a character's gender plays a significant role in the frequency of the action portrayal even when controlling for the yearly trends. These results are show in S5 Table.

In the following section, we discuss the significance of this findings in the context of film theory.

## Discussion

TV and film are among the most universal mass media in history [60]. They have a tremendous power to shape the ways in which people think and behave. When character depictions are perceived by the viewer as similar to their standard everyday reality, the media message is amplified, creating a more powerful and influential suggestion [60, 61].

Studies conducted with small annotated samples of movies and TV shows have suggested that female characters are constantly portrayed to be less dominant, more emotional, less technical, and more nurturing. In contrast, male characters are shown to be assertive, competitive, independent avoiding weakness, insecurities and emotional outbursts [29, 31, 37]. In the following sections, we discuss how our findings on the portrayals of actions can help provide large-scale empirical evidence to corroborate these results. We do so with a focus on character agency and portrayals of emotion.

Additionally, we explore how some of the actions we found to be significantly dependent on the patient's gender can be explained as part of the "male gaze" theory. The 'male gaze' theory [35] posits three different looks associated with a film: one of the camera (usually controlled by a man, either a staffer or director), one for the characters looking at each other, and one originating from the spectators or audiences. In all of these, the woman is the passive receiver of the gaze and the man is the active spectator of the woman; the woman is taken as an object, subjected to a controlling and curious gaze of the man [36]. The results obtained through the presented studies (S1—S4) corroborate that certain actions often associated with gaze typically originate from a male agent, and mostly focus toward a female patient.

While we only present a handful of the results, we make the larger set readily available as a way for researchers across other disciplines to look through it and corroborate with their expertise.

## Characters' agency

Previous works argue that male characters are generally given more agency than female characters [11, 29]. Proportion tests on the difference between the number of portrayals of male agents vs. female agents corroborate this finding. In our dataset, the number of male agent's portrayals significantly higher than that of female agents ($Z = 294.12$, $p < 0.0001$). Moreover, the proportion of female patients is larger than that of male patients ($Z = 294.12$, $p < 0.0001$).

With respect to the actions that significantly depend on the gender of the actor, our results parallel those collected by Sap et al. [29]. Male characters are less likely to be shown 'letting' other male characters do something to them (**S3**: $\beta = -2.27$).

With respect to portrayals across the decades, we see that 'call meeting' is an action that has been consistently portrayed mostly between two male characters (**S4**: $\beta = 8.83$). We hypothesize that this trend reflects the fact that male characters are being shown in professional settings more often than their female counterparts.

## The male gaze theory

Our results highlight differences in the emphasis placed on the female appearance and sexual objectification of women actors. For example, our estimations of actions on the patients (Study 2) suggest that female characters are more likely to be 'gawked' or 'looked at' by other characters ($\beta = -2.76$ and $\beta = -2.88$, respectively). From our study 1, we also found that the action 'stare fascinated' is more frequently portrayed by male agents (**S1**: $\beta = 2.28$). Furthermore, our results of Study 4 (Agent & Patient + Time) suggests that some of these actions are consistent even across decades. After controlling for year of production, male agents are still shown 'looking' significantly more frequent than female agents (**S4**: $\beta = 73.62$), and female patients are significantly more often portrayed as being 'looked' or 'watched' (**S4**: $\beta = -50.26$ and $\beta = -13.87$, respectively).

## Portrayals of emotion

Previous media studies suggested that female characters are typically stereotyped through portrayals of emotional outbursts [37]. In a similar tone, our results highlights that male characters

are statistically less likely to be shown portraying certain actions that convey emotions. For example, **S1: Agent-only** highlights that male characters are less likely to be agents of 'sobbing' and 'crying' (**S1**: $\beta = -1.47$ and $\beta = -1.01$ respectively). Moreover, male characters not often portrayed as baffled or concerned (**S1**: $\beta = -3.72$ and $\beta = -3.27$), nor as agents of snuggling and giggling (**S1**: $\beta = -1.49$ and $\beta = -1.03$). Likewise, our results from **S3: Agent + Patient** suggests that male characters rarely 'scream' at other male characters (**S3**: $\beta = -2.59$); nor do male agents 'laugh' or 'smile' at other male characters (**S3**: $\beta = -2.46$, and $\beta = -2.19$, respectively).

**Violence and aggression.** Media studies have found that women are typically portrayed in distress or need of protection [12, 38, 62, 63]. We have identified particular actions that help corroborate these findings. Across the decades, male agents have been shown to 'get pissed' at other male characters more often (**S4**: $\beta = 6.45$). Other results show that female characters are more likely to be patients of aggressive actions. We see this in examples of actions such as 'kidnap', 'drug', and 'hassle' where the patient is more likely to be a female character (**S2**: $\beta = -2.21$, $\beta = -2.89$, and $\beta = -2.60$ respectively). Similarly, female characters are less likely to be 'lured' by other characters (**S2**: $\beta = -2.41$).

We found an interesting exception to this trend in the action 'shoot'. There is a higher frequency when the target is a male character than when the target is female (**S2**: $\beta = 3.19$). One possible explanation for this result is that male characters just happen to be portrayed more often in action scenes involving a gun. As previous literature suggested, most of the perpetrators are portrayed by middle-aged white male actors [64–66], in action-driven male-dominated narratives where the conflict is resolved with the villain's demise.

**Displays of affection.** Additional results highlight gender differences in the way affective interactions happen on the screen, specifically with respect to the male-to-male pairs. Male characters are often the patient of the 'kiss' (**S2**: $\beta = 3.63$) but not often the initiators of the action (**S1**: 'kiss' $\beta = -2.98$, adjusted p-value $>0.05$). Yet, two male characters are rarely shown on screen 'kissing', 'wrapping [arms]', 'dancing', or 'hugging' (**S3**: $\beta = -3.18$, $\beta = -2.91$, $\beta = -2.91$, and $\beta = -2.78$, respectively).

These results seem to reflect on a societal conception of mixed-sex affection portrayals—what Tillmann [67] calls the 'invisibly ordinary'—where heterosexuality is the predominant assumption for characters in movies. In contrast, the historical notion of same-sex displays of affection is that they evoke disgust, 'cultural squeamishness', and even sometimes, real-life violence [39, 40, 67, 68]. If same-sex shows of affection induce such a response, why is that our results only capture the male-to-male dyad and not the female-to-female dyad? To uncover possible differences in the same-sex dyads, we perform an additional comparison using regression models fitted in the subset where the agent and patient are assigned the same assumed gender (i.e., male-to-male vs. female-to-female). The frequency of the affective actions 'kiss', 'hug' and 'wrap' and the genders of the agent and patient remains significant (t-test, all $p < 0.05$). In all cases, the affective actions were less likely to be portrayed by two males than two females (**same-sex**: $\beta = -3.22$, $\beta = -2.96$, and $\beta = -2.93$ respectively). Thus, our results seem to point towards the sexualization of female shows of affection, specifically in how an on-screen lesbian kiss can be perceived as a form of sexualized entertainment for the heterosexual male viewer [69].

## Conclusion

This work presents a novel large-scale analysis on the actions taken by the characters, and how these actions are related to gender-based stereotypes in media. It is part of an overarching initiative to go beyond simple frequency statistics and assess the quality of character portrayals in media stories told in film and television.

Our results uncover linguistic patterns from the action descriptions in movie scripts where male characters are portrayed with higher agency than female characters; female characters are often cast in emotional supporting roles; male affection is rarely presented on screen, especially when the patient of affection is also a male, and where female characters are often the patients of actions that draw attention to their appearance and looks. Thus, our work complements previous literature and provides large-scale empirical evidence to support their claims.

### Limitations and future work

Some of the limiting factors of our analysis originate from the limitations of the automated processing pipeline. We are bound by the capabilities of the parser to identify words, their part-of-speech tags, as well as the semantic role they play.

Another limitation is due to a nuanced societal and linguistic context that its lost to our analysis. Even when certain words are being used inside the scripts they might communicate a different meaning given what is happening on-screen and the overall narrative of the story. While our large sample of movies, use of highly-contextual embeddings, and aggregation studies aim to coalesce the differences, we can only be certain that we capture but a small sample of the all the possible different linguistic contexts.

Finally, we would like to extend our current framework to incorporate notions of representation intersectionality (e.g., gender, age and race). Even though our current framework provides a way to incorporate these variables, there are still several limitations in automating the labeling of these constructs at a scale (e.g., defining an appropriate ontology).

### Supporting information

**S1 Table. Machine learning model performance.** Complete table of performance results for the SRL systems. Legend: Oracle action denotes models with no automatic action identification. Uncased / Cased refers to the type of pre-trained BERT model used. +fine-tune notes models that were fine-tuned using the manually labeled action description dataset. For LSTM and GRU, size of hidden dimension is given by the number in parenthesis. Best performance highlighted in bold.
(PDF)

**S2 Table. Results for Study 1: Agent-only.** Regression model results for agent's actions. We test the significance of the coefficients through Z-test, and correct for multiple comparisons using the Holm-Bonferroni method. Table shows only significant coefficients with adjusted-$p < 0.05$. Rows are ordered by the magnitude of their coefficient ($\beta$). The direction of the relationship is determined by the sign of the coefficient, with positive coefficients corresponding to actions which are more likely portrayed by a male character. Likewise, negative coefficients present actions that are less likely to be portrayed by male characters. Manually identified errors are color coded (blush—errors due to parsing and lemmatization; gray—errors due to SRL).
(PDF)

**S3 Table. Results for Study 2: Patient-only.** Regression model results for actions done to patients. We test the significance of the coefficients through Z-test, and correct for multiple comparisons using the Holm-Bonferroni method. Table shows only significant coefficients with adjusted-$p < 0.05$. Rows are ordered by the magnitude of the coefficients ($\beta$). The direction of the relationship is determined by the sign of the coefficient, with positive coefficients corresponding to actions more likely done towards male characters. Manually identified errors

are color coded (blush—errors due to parsing and lemmatization; gray—errors due to SRL).
(PDF)

**S4 Table. Results for Study 3: Agent & patient.** Regression model results for the agent–patient interactions. Group encodes gender dynamics (e.g., M→M identify actions done by male characters towards other male characters). We test the significance of the coefficients through Z-test, and correct for multiple comparisons using the Holm-Bonferroni method. Table shows only significant coefficients with adjusted-$p < 0.05$. Rows are ordered by the magnitude of the coefficients ($\beta$). Direction of the relationship is given by the sign and magnitude of $\beta$ with positive values indicating actions more likely portrayed by that group of characters. Manually identified errors are color coded: blush for errors down-streamed from an outside the SRL system (e.g., parsing, lemmatization), and gray for errors due to mislabels coming from our SRL system.
(PDF)

**S5 Table. Results for Study 4: Agent & patient + time.** Regression model results for the agent–patient interactions controlling for year of production. Group encodes gender dynamics (e.g., M→M identify actions done by male characters towards other male characters). A star (*) is used as short-hand for either group (e.g., *→F labels actions where the patient is Female and the agent's gender can be any value). We test the significance of the coefficients through Z-test, and correct for multiple comparisons using the Holm-Bonferroni method. Table shows only significant coefficients with adjusted-$p < 0.05$. Rows are ordered by the magnitude of the coefficient ($\beta$). The direction of the relationship is given by the coefficient's sign, with positive coefficients corresponding to actions more likely portrayed by their encoding group. Manually identified errors are color coded gray for errors due to mislabels coming from our SRL system.
(PDF)

## Acknowledgments

We thank the reviewers for their insightful comments and feedback towards improving this work. VRM would like to acknowledge Prof Jesus Arroyo Relion for his guidance while developing the statistical models. Finally, the authors gratefully acknowledge the members and advisors of USC Center for Computational Media Intelligence.

## Author Contributions

**Conceptualization:** Victor R. Martinez, Krishna Somandepalli.

**Data curation:** Victor R. Martinez.

**Formal analysis:** Victor R. Martinez.

**Funding acquisition:** Shrikanth Narayanan.

**Methodology:** Victor R. Martinez, Krishna Somandepalli.

**Project administration:** Shrikanth Narayanan.

**Supervision:** Shrikanth Narayanan.

**Validation:** Victor R. Martinez.

**Writing – original draft:** Victor R. Martinez.

**Writing – review & editing:** Victor R. Martinez, Krishna Somandepalli, Shrikanth Narayanan.

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
