## [Decision Letter · Decision Letter 0]

13 May 2022

PONE-D-22-04236Boys don’t cry (or kiss or dance): A computational linguistic lens into gendered actions in filmPLOS ONE

Dear Dr. Martinez,

Thank you for submitting your manuscript to PLOS ONE. After careful consideration, we feel that it has merit but does not fully meet PLOS ONE’s publication criteria as it currently stands. Therefore, we invite you to submit a revised version of the manuscript that addresses the points raised during the review process.

We look forward to receiving your revised manuscript.

Kind regards,

Natalia Grabar

Academic Editor

PLOS ONE

Journal Requirements:

2. Please update your submission to use the PLOS LaTeX template. The template and more information on our requirements for LaTeX submissions can be found at http://journals.plos.org/plosone/s/latex

3. Please note that PLOS journals require authors to make all data necessary to replicate their study’s findings publicly available without restriction at the time of publication. Please see our Data Availability policy at https://journals.plos.org/plosone/s/data-availability. As such, please make your full dataset available by either A) uploading the full dataset as supplementary information files, or B) including a URL link in your Data Availability Statement and Methods section to where the full dataset can be accessed

Reviewers' comments:

Reviewer's Responses to Questions

**Comments to the Author**

1. Is the manuscript technically sound, and do the data support the conclusions?

Reviewer #1: Yes

Reviewer #2: Yes

2. Has the statistical analysis been performed appropriately and rigorously? 

Reviewer #1: Yes

Reviewer #2: Yes

3. Have the authors made all data underlying the findings in their manuscript fully available?

Reviewer #1: No

Reviewer #2: Yes

4. Is the manuscript presented in an intelligible fashion and written in standard English?

Reviewer #1: Yes

Reviewer #2: Yes

5. Review Comments to the Author

Reviewer #1: This article proposal presents a large scale experiment aiming at analyzing the way actions in movies are stereotyped according to genders. In order to do so, a simplified version of SRL is performed on a large corpus of scripts, with the identification of actions, agents and patients using BERT+RNN. A throughout statistical analysis is also performed on the results.

The experiment is well-described and sound, the paper is well-written and the results are useful (although not surprising). However, the paper could be improved and I'll try to provide some help with that.

The main design issue with this research is related to the binary classification of gender, which is addressed in a note: "[...] we are limited by the content analysis procedures". This should be more developed and detailed in the limitations and perspectives, so that somebody who would like to do this is a more continuum-oriented way know what to do.

I'm also concerned by the variables being maybe not independent of others which are not taken into account here, like the socio-economic status of the characters. This is highlighted in the limitations concerning race and age, but not the socio-economic status.

The experiment implies a number of layers, it is difficult to assess the quality of the intermediary results (of Spacy, for example). I suggest to add a table showing the different layers and the performance for each of them, if space allows.

Figure 2 is unreadable to me: what is the meaning of the colors? where is the information about female/male characters?

Also, in the huge tables in the Appendix, the best results should be in bold.

Some parts of the paper are a bit too optimistic and should be more nuanced. In particular, Table 2 results are described in the caption as " high precision and recall.", with a recall ranging from 0.63 to 0.8 (precision over 0.9). I suggest to correct this, in order to reflect the results, as the recall cannot be considered "high" here.

The Appendix is longer than the paper itself and contains a lot of interesting information, some of which should appear in the paper itself, in particular concerning the manual annotation process. Concerning the usage of MTurk, the appendix states that the annotators received at least an hourly US minimum wage. Can you explain how and when it was ensured/computed? Which minimum wage (the (very low) federal one?)? Can you also state how many workers (if any) were rejected and for which reason?

"more feminine language": can you be more precise? What does "feminine language" mean in this paper?

Legal issue: the raw corpus is not freely available, instead it is only available for "fair use". This should be cleared in the paper.

I don't think the term "MWE" can be used for expressions like "Elron and Galdalf". These can be considered as named entities, which you do not want to split, but probably not as MWE. Criteria for MWE in UD can be found here: https://universaldependencies.org/workgroups/mwe.html. More details: https://hal.archives-ouvertes.fr/hal-03016721/document

Justification for the answer NO to "Have the authors made all data underlying the findings in their manuscript fully available?":

the details of the results are available in the supplementary materials, but not the manually annotated data and the status of the dataset is not specified in the paper. This should be made clearer.

Finally, 4 references are in fact pre-prints, not peer-reviewed, papers. I suggest to clearly state that they are preprints in the bibliography and to name them "technical reports".

Ref to consider (maybe):

- BUSSO, Lucia ; VIGNOZZI, Gianmarco. Gender Stereotypes in Film Language: A Corpus-Assisted Analysis In: Proceedings of the Fourth Italian Conference on Computational Linguistics CLiC-it 2017: 11-12 December 2017, Rome (https://books.openedition.org/aaccademia/2367?lang=en)

Reviewer #2: First, the "general analysis topic of gender-based portrayals in media" is still essential for organizations' effective and socially responsible communication activities. Thus, the author(s) undertook the task of exploring a complimentary analysis through machine learning based on the character’s actions rather than only dialogue or scene co-appearance better to understand the pervasiveness of harmful gender representations in media. The Manuscript (MS) is interesting and valuable regarding its aim, methodology, and analysis. However, I would like to suggest some revisions;

1. I believe that ıf the structure and flow of MS are revised in an integrative manner, the reader will quickly grasp the idea and follow the process. The current appearance of the MS is an entirely technical paper, whereas the idea/research problem is based on communication and even social problem. The authors also confirm this comment by emphasizing only several analytical/technical contributions of their study on p.4. At the same time, the MS needs to contribute to the literature regarding methodological aspects and conceptual/theoretical aspects. The MS has some potential in this respect.

2. I suggest the author(s) revise the MS by considering the similar headlines below;

- A flow diagram depicting the stages of data collection/ processing stages and analyses step might be handy to catch the reader’s attention and make the readers understand the whole data processing and analysis stage.

- The subtitles under the Conclusion section could be revised by considering the compatibility with the findings in general ( For instance, 4.2 ( The male-gaze theory" does not seem to be the compatible title. The authors use this base or approach just to explain their findings.)

- All of the information in the Appendices are valuable. However, this section seems to be too long as an Appendix. I suggest the authors summarize the quotes from "Materials" to "Experiments" and integrate this information into the main body of the MS. Some of the sections in the Appendices, excluding Tables, are too long and detailed. The authors can put appropriate subtitles If they prefer or necessary. Furthermore, the flow diagram would visually complete the meaning of the whole process.

- the authors state neglecting a time-trend factor in the models as one of the limitations of the study. Unfortunately, this might be an important analytical weakness rather than limitation.At this stage, I would like to ask why do not consider time series based analysis ?

I wish the authors good luck with their research.

6. PLOS authors have the option to publish the peer review history of their article (what does this mean?). If published, this will include your full peer review and any attached files.

Reviewer #1: No

Reviewer #2: No

---

## [Author Response · Author response to Decision Letter 0]

18 Jul 2022

General Response

We thank the reviewers for their thoughtful feedback. We are encouraged by their recognition that the topic is “essential for organization’s effective and socially responsible communication” (R2) and that they found the manuscript and experiments well-written and sound with useful results (R1, R2).

 A common concern among the reviewers was that the appendices contained valuable information that should have been included as part of the main body (R1, R2). We have since summarized the appendices into the paper itself, with a particular focus on the manual annotation process (as suggested by R1). Furthermore, we have followed R2’s suggestion on including a flow diagram in the introduction that we hope better explains the whole data processing and analysis stage.

 Furthermore we have followed the style and grammar suggestions provided by the reviewers. In the following, we try to address the remainder of the reviewers points. 

Response to Reviewer 1:

Q1: The main design issue with this research is related to the binary classification of gender, which is addressed in a note: "[...] we are limited by the content analysis procedures". This should be more developed and detailed in the limitations and perspectives, so that somebody who would like to do this is a more continuum-oriented way know what to do.

Author’s response: We deeply appreciate the reviewer’s comment, however at this time we have our reservations preventing us from venturing any suggestion on what a holistic continuum solution would even look like. While one possibility is to argue that we could represent a character’s expressed gender as a continuum variable, this is still limited by its inherent reliance on a (hetero-)normative onthology that places female–male as polar opposites. We believe that there is still a long-way to go in discussions and implementations before any automatic content analysis procedure can fully capture the nuances and inter-complexity of a character’s (and ultimately a person’s) identity in any comprehensive manner. We believe that these discussions, while still important to have, are out of the scope of our current work.

Q2: I'm also concerned by the variables being maybe not independent of others which are not taken into account here, like the socio-economic status of the characters. This is highlighted in the limitations concerning race and age, but not the socio-economic status.

Author’s response: There might be many hidden factors yet to be considered as explanations for action frequency, yet many of these are unattainable due to a lack of data. Furthermore, since our analysis is performed over a large sample of action descriptions, over a span of 4 decades, and controlling for the factors we do have data for, we suspect that these hidden relationships are marginalized in the aggregative results.

Q3: The experiment implies a number of layers, it is difficult to assess the quality of the intermediary results (of Spacy, for example). I suggest to add a table showing the different layers and the performance for each of them, if space allows.

Author’s response: Our model relies on Spacy only to find the verbs inside the action descriptions. Spacy provides public performance benchmarks on their website (https://spacy.io/usage/facts-figures). Given that we have no reason to believe that the performance of our system would be greatly impacted had we used a different verb identification system, we decided not to include these. We do provide extensive comparisons against other SRL systems (with and without Spacy) in Table S1.

Q4: Figure 2 is unreadable to me: what is the meaning of the colors? where is the information about female/male characters? Also, in the huge tables in the Appendix, the best results should be in bold.

Author’s response: We agree that Figure 2 failed to communicate our original intent and confused the discussion. We have since removed Figure 2 from the updated manuscript. We have also updated the table in the appendix to highlight the best model’s performance in bold.

Q5: Some parts of the paper are a bit too optimistic and should be more nuanced. In particular, Table 2 results are described in the caption as " high precision and recall.", with a recall ranging from 0.63 to 0.8 (precision over 0.9). I suggest to correct this, in order to reflect the results, as the recall cannot be considered "high" here.

Author’s response: We have updated the language used throughout the manuscript in an effort to avoid sounding overly optimistic.

Q6: The Appendix is longer than the paper itself and contains a lot of interesting information, some of which should appear in the paper itself, in particular concerning the manual annotation process. Concerning the usage of MTurk, the appendix states that the annotators received at least an hourly US minimum wage. Can you explain how and when it was ensured/computed? Which minimum wage (the (very low) federal one?)? Can you also state how many workers (if any) were rejected and for which reason?

Author’s response: We have re-structured the manuscript to reflect your suggestions. We have now incorporated the information on dataset creation, methods and experiment (previously found in the appendix) as part of the main body.

We set the remuneration scheme for our task to ensure annotators receive at least minimum wage for their effort. This was calculated for the U.S. as $7.25/hr divided by the expected time it would take to complete a single annotation. Before submitting our task, we did a few dry runs where we estimated the time required to complete a single annotation task. The labeling task (i.e., given an action description, selecting an agent and a patient from a dropdown box) took us about a minute per sample, while the verification task (i.e., deciding whether a proposed label is correct) took half a minute per sample. We have incorporated this response as part of the footnote in page 5.

With respect to the rejection rate, we found that some of the annotators submitted empty results (i.e., did not select a character from the drop-down). We marked their annotations as moot and rejected their tasks. This totalled to 2,162 rejected samples. Additional information can be found in lines 416–423.

Q7: "more feminine language": can you be more precise? What does "feminine language" mean in this paper?

Author’s response: In this context we were referring to gender ladenness. Gender Ladenness, as defined in (Clark and Paivio, 2004) represents the degree of perceived “feminine or masculine association” on a numerical scale ranging from very masculine to very feminine. We have updated the manuscript to make the language clearer.

James M Clark and Allan Paivio. 2004. Extensions of the Paivio, Yuille, and Madigan (1968) norms. Be- havior Research Methods, Instruments, & Comput- ers, 36(3):371–383.

Q8: Legal issue: the raw corpus is not freely available, instead it is only available for "fair use". This should be cleared in the paper.

Author’s response: We appreciate the reviewer’s concern on the legality status of our work, however, assuming that raw corpus refers to ScriptBase, we can assure the reviewer that our use falls well within the limits established by the U.S. Copyright Law section 107 (fair use under non-commercial education and research purposes). If our assumption on what the reviewer meant is wrong, we will be happy to revisit this point.

Q9: I don't think the term "MWE" can be used for expressions like "Elron and Galdalf". These can be considered as named entities, which you do not want to split, but probably not as MWE. Criteria for MWE in UD can be found here: https://universaldependencies.org/workgroups/mwe.html. More details: https://hal.archives-ouvertes.fr/hal-03016721/document

Author’s response: The reviewer is correct in pointing this out. We have updated the manuscript to use the more appropriate term of conjunction.

Q10: Justification for the answer NO to "Have the authors made all data underlying the findings in their manuscript fully available?": the details of the results are available in the supplementary materials, but not the manually annotated data and the status of the dataset is not specified in the paper. This should be made clearer.

Author’s response: We have made all the annotated data and trained models available for anyone to download and use at https://sail/usc/edu/~ccmi/actions-agents-and-patients (Line 109–111).

Q11: Finally, 4 references are in fact pre-prints, not peer-reviewed, papers. I suggest to clearly state that they are preprints in the bibliography and to name them "technical reports".

Author’s response: Thank you for pointing this out. We have updated the manuscript to reflect that these are not peer-reviewed publications.

Response to Reviewer 2:

Q1: A flow diagram depicting the stages of data collection/ processing stages and analyses step might be handy to catch the reader’s attention and make the readers understand the whole data processing and analysis stage.

Author’s response: We thank the reviewer for this suggestion. We have updated the manuscript by including the suggested diagram as Fig 1.

Q2: The subtitles under the Conclusion section could be revised by considering the compatibility with the findings in general ( For instance, 4.2 ( The male-gaze theory" does not seem to be the compatible title. The authors use this base or approach just to explain their findings.)

Author’s response: 

Q3: All of the information in the Appendices are valuable. However, this section seems to be too long as an Appendix. I suggest the authors summarize the quotes from "Materials" to "Experiments" and integrate this information into the main body of the MS. Some of the sections in the Appendices, excluding Tables, are too long and detailed. The authors can put appropriate subtitles If they prefer or necessary. Furthermore, the flow diagram would visually complete the meaning of the whole process.

Author’s response: We have re-structured the manuscript to reflect your suggestions. We have now incorporated the information on dataset creation, methods and experiment (previously found in the appendix) as part of the main body.

Q4: the authors state neglecting a time-trend factor in the models as one of the limitations of the study. Unfortunately, this might be an important analytical weakness rather than limitation.At this stage, I would like to ask why do not consider time series based analysis ?

Author’s response: The reason behind not considering a time-trend as a factor were two fold. First, we did not have complete information on the years of production. Second, as stated, our original models did not converge due to technical limitations of the models–mainly that since they were developed using an old statistical package (based on R) it did not allow for parallel processing. We are happy to let the reviewers know that we have worked out these limitations by collecting additional year-of-release information, and re-writing our statistical models using the more modern Julia programming language. This allowed us to include an additional study (Study 4: Agent & Patients + Time) in which we include a movie’s year-of-release as part of the control variables. The idea behind it is that this model captures the notion that different years have different film trends resulting in different action-frequency distributions. We have updated the manuscript incorporating the details on this time-based study (lines 404–412) and its respective results (lines 503–517) as well as updated the discussion to reflect our gained knowledge on how certain actions are significant even when controlling for yearly trends.

---

## [Decision Letter · Decision Letter 1]

17 Oct 2022

PONE-D-22-04236R1Boys don’t cry (or kiss or dance): A computational linguistic lens into gendered actions in filmPLOS ONE

Dear Dr. Martinez,

Thank you for submitting your manuscript to PLOS ONE. After careful consideration, we feel that it has merit but does not fully meet PLOS ONE’s publication criteria as it currently stands. Therefore, we invite you to submit a revised version of the manuscript that addresses the points raised during the review process.

 Victor, I am again sorry about the delays with the rereviewing.

Thank you for taking into account the previous comments. You have now some additional minor comments to consider.

Thank you for your work.

We look forward to receiving your revised manuscript.

Kind regards,

Natalia Grabar

Academic Editor

PLOS ONE

Journal Requirements:

Reviewers' comments:

Reviewer's Responses to Questions

**Comments to the Author**

1. If the authors have adequately addressed your comments raised in a previous round of review and you feel that this manuscript is now acceptable for publication, you may indicate that here to bypass the “Comments to the Author” section, enter your conflict of interest statement in the “Confidential to Editor” section, and submit your "Accept" recommendation.

Reviewer #3: (No Response)

2. Is the manuscript technically sound, and do the data support the conclusions?

Reviewer #3: Yes

3. Has the statistical analysis been performed appropriately and rigorously? 

Reviewer #3: Yes

4. Have the authors made all data underlying the findings in their manuscript fully available?

Reviewer #3: Yes

5. Is the manuscript presented in an intelligible fashion and written in standard English?

Reviewer #3: Yes

6. Review Comments to the Author

Reviewer #3: The topic is extremely interesting. This is a good paper, and the results are now better explained than in the original submission.

But the tables 5 to 8 in the supplementary material are important, and they are still not easy to understand. Why are the rows ordered by Z, wouldn't ordering by Estimate make more sense? It's not clear what the column "significance" is based on, or what information is adds. I see entries with two stars with a larger Estimate than entries with three stars. For tables 5 and 6 (agent-only and patient-only) it would be nice to have an easy way to see the top actions for males and the top actions for female separately. Either color code them differently, or put them in different tables, or sort the rows to have all the male terms together and all the female terms together...

Other, more minor points, that are not an obstacle to accept the paper:

Figure 1: Great figure, it helps a lot to guide the reader. Small layout issue for the lines connected to the boxes agents/actions/patients : The lines don't bend where you meant it to bend, so it looks like actions and patients are connected to each other, but not to annotation on the left.

line 172~181 verification steps of the annotations: So you have 3 annotators per action, then a fourth annotator to verify the results, then one of the author looks at it one more time. Why all these different steps? Did you have a lot of bad annotations using just the 3 initial annotators agreement? It would be a plus to have this explained in a few words.

7. PLOS authors have the option to publish the peer review history of their article (what does this mean?). If published, this will include your full peer review and any attached files.

Reviewer #3: **Yes: **Sam Bigeard

---

## [Author Response · Author response to Decision Letter 1]

14 Nov 2022

November 12th, 2022

PLOS One Editorial Team,

Dear Editorial Team,

We would like to thank the reviewer for their effort in providing insightful feedback to improve our work. Throughout this process, we continue to be encouraged by the reviewer’s recognition of the interesting topic, and the value that our work provides to the discussion on gender bias in the media.

While working on this revision, we have identified and addressed a limitation in our statistical analysis that resulted in an overestimation of the number of significant coefficients. In summary, our Z-tests for coefficient significance (H0: \\beta = 0 vs. H1: \\beta != 0) were not controlled for multiple comparison errors (also known as familywise errors). To address this limitation, we incorporated a post-hoc Holm-Bonferroni correction for all GLME models. In other words, we are employing a higher standard for what we consider to be a significant result. The presented revision collects only the results that are found to be significant after correction (i.e., adj-p < 0.05). 

We have updated the results, tables and discussion section to contextualize only those results that meet the new (and more appropriate) standard. This resulted in the removal of certain citations, as these are no longer needed. 

##### Response to Reviewer 3:

Comment regarding the tables 5 to 8: We agree with the reviewers comment that the tables are hard to parse and not easy to understand. In addition to increasing the standard for what we consider a significant coefficient, we improve the readability of the tables in the supplementary materials as follows:

 * We removed the significance column. This column aimed to provide an easy way to discern the power of the test (i.e., how likely is this coefficient to be different from zero?). However, after correcting for multiple comparisons, the coefficients deemed significant were those with the smallest p-values (most of which had adj-p = 0) which resulted in all having the same significance level. 

 * We followed R3’s suggestion on presenting the coefficients ordered by their magnitude.

 * Additionally, we followed R3’s suggestion to split the tables into sub-tables according to the direction of the relationship (ie. less or more likely to be portrayed by X gender). This direction is given by the coefficient sign. We caption sub-tables with a brief explanation of the interpretation for each of the results presented (e.g., “Actions where the agent is more likely to be male”).

Figure 1: We updated the figure to fix the layout issue. (Thanks for pointing this out). 

Verification steps of the annotations: We have included a paragraph with a brief explanation of why we decided on a two-step verification process. In summary, we had to be certain that the labels met a minimum quality standard since the models we are using make no distinction between subjects (i.e., person, animal or thing that is in the receiving end of the action) and patients (ie. characters receiving actions).

In addition to the above comments, we have corrected additional spelling and grammatical errors.

We look forward to hearing from you in due time regarding our submission and to respond to any further questions or comments you may have.

Sincerely,

Dr. Victor R Martinez

Corresponding Author

---

## [Editor Report · Decision Letter 2]

21 Nov 2022

Boys don’t cry (or kiss or dance): A computational linguistic lens into gendered actions in film

PONE-D-22-04236R2

Dear Dr. Martinez,

We’re pleased to inform you that your manuscript has been judged scientifically suitable for publication and will be formally accepted for publication once it meets all outstanding technical requirements.

Kind regards,

Natalia Grabar

Academic Editor

PLOS ONE

Additional Editor Comments (optional):

The comments of the reviewers have been taken into account. The Authors also corrected some previous limitations of their methodology. This improved the overall quality of the submission, which is acceptable for publication now.

---

## [Editor Report · Acceptance letter]

23 Nov 2022

PONE-D-22-04236R2 

Boys don’t cry (or kiss or dance):
A computational linguistic lens into gendered actions in film 

Dear Dr. Martinez:

I'm pleased to inform you that your manuscript has been deemed suitable for publication in PLOS ONE. Congratulations! Your manuscript is now with our production department. 

Kind regards, 

on behalf of

Dr. Natalia Grabar 

Academic Editor

PLOS ONE